# Unveiling New Druggable Pockets in Influenza Non-Structural Protein 1: NS1–Host Interactions as Antiviral Targets for Flu

**DOI:** 10.3390/ijms24032977

**Published:** 2023-02-03

**Authors:** Andreia E. S. Cunha, Rui J. S. Loureiro, Carlos J. V. Simões, Rui M. M. Brito

**Affiliations:** 1Coimbra Chemistry Center—Institute of Molecular Sciences (CQC-IMS), Department of Chemistry, University of Coimbra, 3004-535 Coimbra, Portugal; 2BSIM Therapeutics, Instituto Pedro Nunes, 3030-199 Coimbra, Portugal

**Keywords:** influenza, NS1, effector domain, molecular dynamics, druggability, protein–protein interactions, PI3K, TRIM25

## Abstract

Influenza viruses are responsible for significant morbidity and mortality worldwide in winter seasonal outbreaks and in flu pandemics. Influenza viruses have a high rate of evolution, requiring annual vaccine updates and severely diminishing the effectiveness of the available antivirals. Identifying novel viral targets and developing new effective antivirals is an urgent need. One of the most promising new targets for influenza antiviral therapy is non-structural protein 1 (NS1), a highly conserved protein exclusively expressed in virus-infected cells that mediates essential functions in virus replication and pathogenesis. Interaction of NS1 with the host proteins PI3K and TRIM25 is paramount for NS1’s role in infection and pathogenesis by promoting viral replication through the inhibition of apoptosis and suppressing interferon production, respectively. We, therefore, conducted an analysis of the druggability of this viral protein by performing molecular dynamics simulations on full-length NS1 coupled with ligand pocket detection. We identified several druggable pockets that are partially conserved throughout most of the simulation time. Moreover, we found out that some of these druggable pockets co-localize with the most stable binding regions of the protein–protein interaction (PPI) sites of NS1 with PI3K and TRIM25, which suggests that these NS1 druggable pockets are promising new targets for antiviral development.

## 1. Introduction

Influenza (flu) is an acute contagious respiratory illness resulting from infection with influenza viruses. Influenza infections represent a serious public health threat worldwide, with winter seasonal outbreaks being responsible for 1 billion cases, 3 to 5 million severe cases, and 290,000 to 650,000 deaths each year [1,2]. Moreover, the influenza virus poses a permanent risk of the emergence of severe flu pandemics, which, as stated by the WHO, are a permanent threat to become a devastating global health event [2]. Besides its serious health impacts, influenza infections also represent a significant economic burden [3], estimated at 11.2 billion dollars yearly in the U.S. [4] and 6 to 14 billion euros yearly in the EU [5].

Influenza’s most common clinical manifestations include fever, cough, sore throat, rhinorrhoea, headache, myalgia, and weakness, and, despite being usually mild in healthy adults, they can be severe and fatal to high-risk groups such as the elderly, children under 59 months and patients with chronic diseases or immunosuppression [6,7,8]. Moreover, healthy adults are also at risk during pandemic outbreaks [9].

Influenza viruses belong to the family Orthomyxoviridae, have a negative sense eight-segment single-stranded RNA genome encoding 11 proteins, and are divided into types A, B, C, and D, of which only A, B, and C infect humans (reviewed in [10]). Influenza viruses A and B are pleomorphic, generally presenting a spherical shape and sizes in the range of 80–120 nm, although cord-like shapes can also occur [11]. Influenza C viruses usually have a filamentous shape with sizes reaching 500 nm [11]. Human influenza A and B viruses are responsible for seasonal flu outbreaks, while influenza C usually causes mild infections. Influenza A is the only type of influenza virus responsible for the emergence of flu pandemics and is the most pathogenic for the human population, being, therefore, the main target of influenza vaccines. Type A influenza viruses are divided into multiple subtypes based on the combination of two proteins on the viral surface: hemagglutinin (HA) and neuraminidase (NA), which have, respectively, 18 and 11 subtypes.

Influenza viruses have a high rate of evolution that arises from two main processes: antigenic drift and antigenic shift (reviewed in [10,12]). Antigenic drift is the gradual accumulation of changes in the viral genome over time that originates a virus of the same subtype but is unrecognizable by the host immune system and results from the lack of proofreading ability in RNA viruses. Antigenic shift or reassortment occurs in influenza A viruses (IAV) and is characterized by the mixture and rearrangement of RNA segments from different IAV subtypes infecting the same cell, originating a new subtype for which most of the population does not have preexisting immunity. This process leads to new pandemics when the newly generated subtype is successful at human-to-human transmission [13]. The diversity of virus strains and subtypes arising from these two processes significantly reduces vaccine effectiveness and leads to the yearly update of vaccines accordingly to the strains circulating each flu season [14].

Besides vaccination, the other main strategy to counteract influenza infections is the use of antiviral drugs. However, the high rate of mutation of the virus has led to increasing levels of drug resistance and therapeutic inefficacy [11,15]. Indeed, there is currently only one class of drugs (i.e., neuraminidase inhibitors) being used in the clinic [8,11], which makes the development of new antivirals against novel viral biological targets an urgent need.

One of the most promising new targets for influenza antiviral therapy is the non-structural protein 1 (NS1), a highly conserved protein exclusively expressed in virus-infected cells that mediates essential functions in virus replication and pathogenesis [10,16]. NS1 role in influenza infection results from two main processes: promotion of viral replication and evasion of the host immune system [16]. Several studies [17,18] have shown that influenza viruses without NS1 suffer a decrease in their replication and dissemination, which suggests that the development of new therapies targeting this viral protein could improve the efficacy and outcomes of influenza antiviral treatment. 

The NS1 structure comprises an RNA-binding domain (RBD), responsible for binding the viral RNA, connected through a variable length flexible linker to the effector domain (ED), responsible for interacting with multiple host proteins, and a disordered C-terminal tail [10]. NS1 has a high structural plasticity, which allows it to establish multiple functionally relevant protein–protein interactions (PPI) with host cell proteins, mainly through the ED domain, such as with CPSF30 [17], TRIM25 [19] and PI3K [20].

NS1 interaction with PI3K is of paramount importance in the early stages of infection. PI3K is a lipid kinase involved in the PI3K/Akt signaling pathway, which regulates key cellular processes such as cell survival, apoptosis, proliferation, and differentiation and exerts pro-survival and antiapoptotic effects [21,22]. The activation of PI3K induced by NS1 binding leads to the inhibition of apoptosis in influenza virus-infected cells, thus promoting viral replication [23,24,25]. Additionally, this interaction could also have other cellular effects dependent on the PI3K spatial distribution and on the activation of distinct downstream pathways [26,27]. 

NS1 interaction with TRIM25 also plays an essential role in influenza infection and pathogenesis. TRIM25 is a member of the TRIM protein family of RING-type E3 ligases that play crucial roles in the regulation of the innate immune signaling pathways responsible for the induction of the type I interferon (IFN) response and for the production of inflammatory cytokines [19,28,29]. Specifically, upon the interaction of the viral RNA sensing RIG-I protein with viral RNA, TRIM25 poly-ubiquitinates the second CARD domain of RIG-I, leading to the activation of the downstream effector mitochondrial antiviral signaling (MAVS) protein and to the subsequent production of interferon α/β [30,31]. The binding of NS1 to the coiled-coil domain of TRIM25 prevents the correct positioning of the PRYSPY domain of TRIM25 required for substrate ubiquitination, hence suppressing the RIG-I/MAVS signaling pathway and interferon production [19]. Additionally, it was recently reported that NS1 also inhibits the TRIM25-mediated ubiquitination of another viral sensing protein, DDX3X, leading to the suppression of IFN production [32], raising the possibility of other effects of the NS1-TRIM25 binding besides the blockage of RIG-I ubiquitination.

Considering the crucial role of the NS1 interactions with PI3K and TRIM25 on influenza infection and pathogenesis and the fact that the interaction surfaces of NS1 with both proteins overlap, in the current work, we explored the potential of these two PPIs to be targets of new antiviral agents.

With the help of in silico methodologies and advances in hardware and computer performance, rational drug discovery has advanced at a breakneck pace. While still in its infancy in comparison to traditional drug discovery methodologies, computer-aided drug discovery (CADD) has aided in the delivery of several success stories [33,34,35,36] that have instilled confidence in its continued use in the pharmaceutical sector. Before entering detailed experimental assays, CADD can save time and money by identifying probable hits for a therapeutic target [37].

A thorough effort on target validation and characterization is required before beginning a drug discovery effort on a new biological target, such as a protein [38]. Because proteins are dynamic and can undergo multiple conformational changes, one important aspect of structure-based ligand design is to assess the protein conformation dynamics of the biological target. Not only major conformational changes but also minor conformational adjustments involving residue side chain movements can impact the complementarity between the ligand and the protein’s binding site. Because protein flexibility influences the variety of possible target conformational states for ligand binding, MD simulations can help in drug design by providing information on the target’s dynamic character [39].

Despite significant progress in the experimental and computational approaches used in the drug discovery process over the years, a remarkable overall failure rate of about 96% has been recorded in various drug discovery projects due to the “undruggability” of various identified protein targets among other challenges [40,41,42]. The target “druggability” (the ability of a biological target to be modulated through binding to a drug-like molecule) is one important feature to be evaluated at the early stages of the drug discovery process. 

The goal of the present study was to evaluate the druggability of NS1 by considering the structure’s intrinsic flexibility. Using the three-dimensional structure of full-length NS1, we investigated the structure’s stability and flexibility using multiple molecular dynamics (MD) simulations. Additionally, we looked at the structure’s ability to generate potential binding pockets for small molecules, specifically in the regions of PI3K and TRIM25 binding interfaces. We also calculated the druggability potential of the conserved pockets, or their ability to bind drug-like ligands, as determined by Fpocket [43]. This allowed us to identify promising druggable pockets and to confirm the potential of the NS1’s effector domain (ED) as a drug target.

## 2. Results and Discussion

### 2.1. NS1 Structure’s Dynamics

NS1 is preferentially a homodimer with 230 residues per polypeptide chain organized in two structural domains: the RNA-binding domain (RBD) and the effector domain (ED) (Figure 1). The RBD is formed by residues 1–73 and in solution tends to dimerize. Each RBD chain is linked to the effector domain (residues 81–230) by a flexible linker region. As a result of the linker length and structural flexibility, the orientation of the two effector domains relative to each other and to the RBD may vary significantly [44,45].

Although it is well known that NS1 has high conformational plasticity with concerns to the spatial relation between the RBD (RNA binding) and ED (effector) structural domains, our goal in the present work is to access the potential presence of stable and/or transient pockets at the surface of the protein that could be exploited as binding sites for small molecule modulators of NS1 activity. With this objective in mind, we carried out a series of relatively short (100 ns) molecular dynamics (MD) simulations of full-length NS1 and two molecular complexes of NS1: (i) ED-NS1 with the iSH2 domain of the human p85β subunit of PI3K; and (ii) NS1 with the coiled-coil (CC) domain of human TRIM25 (for more details, see Section 3).

To evaluate the backbone stability and conformational fluctuations of NS1’s structural domains, six replicas of 100 ns molecular dynamics (MD) simulations of the full-length (FL) NS1 were run under the physiologically relevant conditions of 310 K and pH 7.0. The root mean square fluctuation (RMSF) of backbone Cα atoms for all the simulations was computed and plotted (Figure 2), showing that both domains of NS1 are very stable during the molecular dynamics simulations, and even more so the effector domain. The NS1 regions of higher conformational variability are the N-terminal region (RBD) and the linker region. In the case of RBD, the region with higher flexibility occurs in an inter-helical loop between residues 24 and 31, while for the ED, the regions with higher flexibility are those in the vicinity of residues 138, 153, and 165, which are all loop regions. It is thus clear that, if considered individually, each one of the two globular domains of NS1 does not suffer major conformational changes.

### 2.2. Pocket Identification and Analysis

To assess the potential of NS1 as a drug target for new influenza antivirals, we evaluated the formation of stable or transient druggable pockets on the surface of NS1 along the MD trajectories using the Fpocket algorithm, as described in Section 3.4. A druggability score was assigned to each pocket identified on the NS1 surface (Figure 3). The druggability score ranges between 0 and 1, with 1 meaning that drug-like small molecules can bind to the identified pocket with a high probability and 0 meaning that the pocket is non-druggable or unlikely to interact with a drug-like small molecule.

As can be seen in Figure 3, there is a small number of druggable pockets per conformation sampled (each vertical line in the graphs of Figure 3), and these pockets fluctuate between druggable and non-druggable states in consecutive protein conformations sampled at one-nanosecond intervals. We selected the pockets with a druggability score above 0.5, the minimum consensus cutoff for a pocket to be considered druggable, to analyze the residues comprising the pockets and the associated molecular descriptors.

Fpocket calculates the druggability score of a pocket using five different descriptors: (i) the number of alpha spheres fitting the pocket, (ii) the density of the cavity, (iii) the polarity score, (iv) the mean local hydrophobic density, and (v) the proportion of apolar alpha spheres fitting the pocket. Using principal component analysis (PCA), we examined the relationship between the druggability score and the five descriptors. The descriptors were projected onto the first plane of a PCA, which captures 87.64% of the variability (Figure 4). The PCA of the five descriptors reveals that the number of alpha spheres fitting the pocket and the mean local hydrophobic density contribute the most to the druggability score. The number of alpha spheres is related to the size of the cavity, and the mean hydrophobic density identifies the hydrophobic character of the pocket.

For a more in-depth analysis of the pockets, we analyzed the residues belonging to the pockets with the highest druggability score (Figure 5). The residues belonging to the pocket identified in Figure 5a are R88, Y89, L95, M98, S135, V136, I137, E142, T143, and I145, and the residues belonging to the pocket identified in Figure 5b are A86, S87, S135, E142, T143, L144, I145, and P162.

The residues in the interface of the ED-NS1 with PI3K described in the literature are S87, R88, Y89, T91, M93, L95, E96, E97, M98, S99, R100, D101, W102, C116, R118, N133, F134, S135, E142, T143, L144, I145, L146, R148, E159, S161, P162, L163, P164, and S165 [24,26,46,47,48]. Most of the residues identified in the pockets are located in the interaction surfaces with PI3K and TRIM25 or in their proximity.

### 2.3. Characterization of Interactions between Influenza NS1 and Human PI3K or TRIM25

We conducted classical MD simulations of the complex between the ED domain of NS1 from the 1918 pandemic H1N1 influenza A virus and the iSH2 domain of the p85β subunit of human PI3K (PDB ID: 6OX7) and analyzed its structural stability by computing the root mean square deviation (RMSD) for both the complex and the interfacial region as well as the root mean square fluctuation (RMSF) of the interfacial region for all frames of the trajectories. The same procedure was applied for the complex between NS1 from the A/Puerto Rico/8/1934 strain of H1N1 influenza A virus and the coiled-coil (CC) domain of human TRIM25.

The Cα-RMSD for the complex between ED-NS1 and PI3K (Figure 6) shows a very stable structure, presenting low and constant RMSD values along the three trajectories. Indeed, the average RMSD values are 4.1 ± 0.9 Å in replica 1, 3.9 ± 0.6 Å in replica 2, and 3.7 ± 0.9 Å in replica 3.

A similar trend is observed in the complex between NS1 and TRIM25, with the average RMSD over the Cα atoms ranging from 3.3 ± 0.6 Å in replica 2 to 3.6 ± 0.7 Å in replica 1 (Figure 7).

To study in more detail the protein–protein interfacial region for each of the complexes, we defined a set of residues that fulfill the criteria described in Section 3.3 and correspond to 30 NS1 residues and 20 PI3K residues in the case of the NS1-PI3K complex (Table 1) and to 22 NS1 residues and 24 TRIM25 residues in the case of the NS1-TRIM25 complex (Table 2). These are essential regions both to determine the stability of the protein–protein interactions and from a drug design perspective. In both cases, the low RMSD values indicate globally high stability of the interfaces (Figure 8a,b and Figure 9a,b). Nevertheless, a relatively small increase in RMSD (of about 1 Å) is observed for the NS1 and TRIM25 interfacial residues between the 40 and 50 ns of simulation, which denotes that the interface still has some local flexibility despite its overall stability. Indeed, the average RMSD of the interface of both NS1 and PI3K varies from 2.0 ± 0.2 Å in replica 1 to 2.6 ± 0.3 Å in replica 2, while the average RMSD of the interaction surface of NS1 varies between 1.8 ± 0.3 Å in replica 1 and 2.3 ± 0.2 Å in replica 2. Similarly, the average RMSD of the interface of both NS1 and TRIM25 varies from 2.0 ± 0.2 Å in replica 2 to 2.1 ± 0.4 Å in replica 1, while the average RMSD of the interaction surface of NS1 varies between 1.4 ± 0.1 Å in replica 2 and 1.7 ± 0.3 Å in replica 1.

Furthermore, the RMSF values of the interfacial residues of NS1 in the NS1-PI3K and NS1-TRIM25 complexes show very low flexibility, with average values ranging from 1.0 ± 0.5 Å in replica 1 to 1.1 ± 0.5 Å in replica 3 for the NS1-PI3K complex (Figure 10a) and from 0.8 ± 0.4 Å in replica 2 to 0.9 ± 0.5 Å in replica 1 for the NS1-TRIM25 complex (Figure 10b). Additionally, most of the NS1 interfacial residues on the two complexes present RMSF values lower than 1.0 Å, which further indicates the high stability of the interaction surfaces.

The analysis of conformational fluctuations in the interfacial regions of the complexes NS1/PI3K and NS1/TRIM25 showed very stable interfaces over the course of the MD simulations. The stability of the interactions between influenza’s NS1 protein and the human PI3K and TRIM25 proteins, along with their recognized biological significance in the mechanisms of infection and pathogenesis, suggest that they may be suitable drug targets for new influenza antivirals if they co-localize with druggable pockets.

### 2.4. Binding of NS1 to PI3K and TRIM25 Is Mostly Driven by the Same Six Hotspot Residues

To further explore the type and most prevalent molecular interactions involved in influenza’s NS1 binding to its molecular partners PI3K and TRIM25, we conducted a fine-grained description, over the simulation time, of the preservation of the interfacial pairwise residue–residue contacts and identified the residues that participate to a larger extent on highly conserved residue–residue contacts. This analysis is essential to identify the main inter-residue contacts at the molecular complexes interfaces and the essential residues in complex formation and stabilization (i.e., interaction hotspots).

The analysis of the preservation of the pairwise residue–residue contacts at the interfaces was performed with the MDCons standalone tool [49] and, considering the similarity of the trajectories for each complex and for the sake of simplicity, was restricted to one trajectory per complex. The criterion used to define a residue–residue interfacial contact considers that two interfacial residues are in contact if any of the atoms of one residue is within 5 Å of any atom of the second residue in the interaction partner (the criteria used in CAPRI [50]). In the case of the NS1-PI3K complex, the analysis indicated that 19 inter-residue contacts are completely conserved during the 100 ns of simulation, and the other 29 contacts have more than 70% conservation (Table 3 and Figure 11a), which corresponds to a proportion of 67.74% of the total number of inter-residue contacts in all frames, underlining the stability of the interface. The highly conserved interfacial contacts cluster mainly in the region 87–101, followed by smaller clusters in regions 133–135, 142–148, and 161–163. 

Concerning the NS1-TRIM25 complex, we observed that 33 inter-residue contacts are completely conserved during the 100 ns of simulation, and 53 have more than 70% conservation (Figure 11b,c and Table 4 and Table 5), corresponding to proportions of 59.02% and 62.50% of the total number of inter-residue contacts involving the chains I and N of TRIM25 in all frames, respectively, underlining the stability of the interface. The highly conserved interfacial contacts cluster mainly in the region 87–101, followed by smaller clusters in regions 133–135, 145–148, and 161–164.

The number of highly conserved interfacial contacts (i.e., contacts with >70% residence time) each residue participates in was computed (Table 6 and Table 7), determining which residues are more often involved in highly conserved interface contacts and thus likely essential in complex formation and stabilization. This analysis indicates that 66% and 96% of the interfacial residues of NS1 participate in highly conserved interfacial contacts with PI3K and TRIM25, respectively, and pointed out Tyr 89, Leu 95, Met 98, Ser 99, Asp 101 (Glu 101 on the NS1-TRIM25 complex), and Ile 145 as key residues in the stabilization of the two complexes, hence called interaction hotspots. Tyr 89 is a highly conserved residue in human influenza A virus [20,51] and has been recognized in several experimental studies as an essential residue in the interaction between NS1 and PI3K by establishing a particularly strong hydrogen bond with Asp 575 of PI3K [20,46,51,52,53]. This hydrogen bond is buried in the hydrophobic cluster of the binding interface and is preserved in 76.22% of our MD simulation, a much higher value than any other interfacial hydrogen bond (Table 8). However, residues S87, E96, S99, D101, and P162 also participate in relatively stable hydrogen bonds (Table 8). These results are in very good agreement with recent experimental data based on alanine scanning mutagenesis and biolayer interferometry, which showed that besides Tyr 89, residues Leu 95, Met 98, Ser 99, Asp 101, and Ile 145 are essential in NS1 binding to PI3K [53].

The goal of this study was to assess the druggable potential of NS1’s effector domain. By relying only on rigid crystallographic structures, one may miss important conformation fluctuations that are sampled over time, including ones that can be targeted by small drug-like compounds. Pockets at the surface of a protein can transiently form as a result of local flexibility, and it is important to evaluate their physicochemical and geometric properties to predict their druggability and their capacity to accommodate a drug-like molecule. At the same time, we wanted to evaluate the stability and flexibility of the interface between NS1’s effector domain and host proteins, such as PI3K and TRIM25, in order to identify the critical intermolecular contacts and residues (interaction hotspots) establishing those fundamental molecular interactions in influenza infection.

From the work presented here on pocket identification at the surface of NS1’s effector domain, their druggability, and identification of the main residues mediating the interactions between NS1 and two of its molecular patterns (PI3K and TRIM25), we concluded on the existence of six interaction hotspot residues (Tyr 89, Leu 95, Met 98, Ser 99, Asp 101 or Glu 101, and Ile 145), which are all located in or near druggable pockets, denoting a high likelihood that a drug-like molecule could bind to these pockets and disrupt these essential PPIs. This suggests that these druggable pockets may be suitable drug targets for the development of new effective antivirals against influenza infections.

## 3. Materials and Methods

### 3.1. NS1 Three-Dimensional Protein Structures

We used the X-ray crystal structure of full-length H6N6 NS1 (PDB ID: 4OPH) as a starting point for molecular dynamics (MD) simulations. The recombinant protein used in the crystal structure determination was expressed in *E.coli*, and its amino acid sequence corresponds to the avian influenza A H6N6 virus (A/blue-winged teal/MN/993/1980) [45]. Additionally, the recombinant protein harbors the R38A-K41A double substitution designed to prevent aggregation of the full-length protein [45].

We also conducted classical molecular dynamics (MD) simulations of two NS1 molecular complexes starting from the following crystallographic structures: (i) the complex between the ED domain of NS1 from influenza A H1N1 virus (A/Brevig Mission/1/1918) and the iSH2 domain of the human p85β subunit of PI3K (PDB ID: 6OX7) [20] (Figure 12a); and (ii) the complex between NS1 from influenza A H1N1 virus (A/Puerto Rico/8/1934) and the coiled-coil (CC) domain of human TRIM25 (PDB ID: 5NT2) [19] (Figure 12b) to evaluate the stability of the complexes and particularly of their protein–protein interfaces.

### 3.2. Molecular Dynamics (MD) Protocol and Analysis

For the MD simulations of full-length NS1, we started by reversing the R38A-K41A double substitution present in the X-ray 3D structure to its wild-type sequence R38-K41 using the Rotamers tool and the Dunbrack library in CHIMERA (https://www.cgl.ucsf.edu/chimera/; version 1.16, University of California, San Francisco, CA, USA) [54,55]. Then, CHIMERA was used to assign amino acid residue protonation states at pH 7 and to build missing side chains, ensuring the absence of any atom clashes. MD simulations were carried out with Gromacs v. 2016.4 [56], using the Amber99SB-ILDN [57] force field under periodic boundary conditions and with an explicit water box of TIP3P water molecules (14 Å thick layer of water). Non-bonded interactions were truncated at a cutoff distance of 10 Å for the electrostatic twin-range cutoff and the Van der Waals cutoff. The particle mesh Ewald (PME) method [58] was used for the evaluation of long-range electrostatic interactions. The energy of the system was minimized over a maximum of 50,000 cycles using the steepest descent algorithm. Each production MD simulation was preceded by a 200 ps NVT (constant number of particles, volume, and temperature) equilibration followed by a 200 ps NPT (constant number of particles, pressure, and temperature) equilibration, during which harmonic restraints were imposed on the atomic positions of the protein. The pressure was treated with the Parrinello–Rahman algorithm [59] (1 bar with a coupling constant of 2.0 and isothermal compressibility of 4.5 × 10^−5^ bar^−1^). The temperature was treated with the v-rescale algorithm [60] (310 K with a coupling constant of 0.1), with solute and solvent being separately coupled to the temperature bath. The LINCS algorithm was applied to all bond lengths to constrain them, which allowed the use of an integration time step of 1 fs. Six independent MD simulation replicas of 100 ns were performed in rhombic dodecahedral boxes with periodic boundary conditions and solvation with 52,852 TIP3P water molecules to ensure that each periodic image of a protein only interacts with itself and not to the other periodic images. MD simulations were visualized using Visual Molecular Dynamics 1.9.4 (https://www.ks.uiuc.edu/Research/vmd/; version 1.9.4, University of Illinois, Urbana, IL, USA) [61] and PyMOL (http://www.pymol.org; version 2.5, Schrödinger, New York, NY, USA) [62]. Using GROMACS tools, several properties were calculated along the simulations to validate their quality and stability, including root mean square deviation (RMSD), root mean square fluctuation (RMSF), and gyration radius.

### 3.3. Molecular Dynamics Simulations of the NS1-PI3K and NS1-TRIM25 Complexes

To study the stability and to identify the main interactions mediating some of NS1’s critical protein–protein interactions (PPIs), we conducted classical molecular dynamics (MD) simulations using as starting points the following crystallographic structures: (i) the complex between the ED domain of NS1 from influenza A H1N1 virus (A/Brevig Mission/1/1918) and the iSH2 domain of the human p85β subunit of PI3K (PDB ID: 6OX7) [20], and (ii) the complex between NS1 from influenza A H1N1 virus (A/Puerto Rico/8/1934) and the coiled-coil (CC) domain of human TRIM25 (PDB ID: 5NT2). In the case of the NS1-TRIM25 complex, comprising one NS1 dimer and two TRIM25-CC dimers, we reduced the system to simulate only one NS1 monomer (chain D) complexed to one TRIM25-CC dimer (chains I and N), considering the large size of the initial structure and that the two interfaces between NS1 and TRIM25 are identical.

The simulations were performed with the GROMACS v.2016.4 software suite [63,64] using the Amber ff99SB-ILDN force field [57]. For the NS1-PI3K complex, three replicas of 100 ns each were performed in rhombic dodecahedral boxes with periodic boundary conditions and solvation with 74,942 TIP3P water molecules (i.e., 25 Å-thick layer of water) to ensure that each protein only interacts with its complex partner in one direction. The same settings were applied to the NS1-TRIM25 complex except for the number of replicas (i.e., two) and the number of solvent molecules (i.e., 209,064 correspondents to a 12 Å-thick layer of water). The leap-frog integrator algorithm was used with a 2 fs time step. The non-bonded interactions were treated with a 10 Å twin-range cutoff and updating the neighbor list every 40 fs. The long-range electrostatic interactions were treated using the particle mesh Ewald (PME) method [58] with an interpolation order of 4 and a grid spacing of 0.16 nm. The temperature was treated with the v-rescale algorithm [60] (300 K with a coupling constant of 0.1), with solute and solvent being separately coupled to the temperature bath. The pressure was treated with the Parrinello–Rahman algorithm [59] (1 bar with a coupling constant of 2.0 and isothermal compressibility of 4.5 × 10^−5^ bar^−1^). The LINCS algorithm was used to constrain all bonds in the NS1-PI3K complex and only those involving hydrogen atoms in the NS1-TRIM25 complex. MD trajectory analysis was conducted using built-in GROMACS tools as well the MDCons standalone tool (version 2.0 of January 2015, KAUST—University of Napoli “Parthenope”, Naples, Italy) [49] (calculation of interfacial contacts conservation during the simulation), the VMD HBonds Plugin (https://www.ks.uiuc.edu/Research/vmd/plugins/hbonds/; version 1.9.4a53 of June 29, 2021, University of Illinois, Urbana, IL, USA, with a donor-acceptor distance cutoff of 3.2 Å and the default value for the angle cutoff) (calculation of hydrogen bonds conservation during the simulation) and others developed in-house (available at https://github.com/RuiJoaoLoureiro2019/PPI-MD-Analysis). Structure and trajectory visualization was performed with PyMol (http://www.pymol.org; version 0.98 of January 2005, Schrödinger, New York, NY, USA).

The definition of protein–protein interface used in the analysis of the simulations was based on the fulfillment of at least one of two criteria for a given residue: (i) any of the residue’s constituent atoms are within 5 Å of any atom in the interaction partner (the criteria used in the Critical Assessment of Predicted Interactions (CAPRI) [50] and in the MDCons tool); and (ii) difference in area ≥1 Å^2^ of the residue’s solvent accessible surface between the bound and unbound states (criteria used in the PyMol script to identify interfacial residues InterfaceResidues.py).

### 3.4. Druggable Pockets Identification and Analysis

To evaluate the formation of pockets on the surface of NS1, protein conformations along the MD trajectories were sampled at one-nanosecond intervals. We estimated NS1 surface pockets using Fpocket (https://github.com/Discngine/fpocket; version 3.0, Discngine, Paris, France), a geometry-based method that investigates all cavities of a protein independently of any ligand information [43].

In the present study, we excluded the smallest pockets, characterized by a volume smaller than 500 Å^3^, which is the minimum size appropriate to bind drug-like molecules [65]. Pockets were described using molecular descriptors, which include (i) physicochemical properties such as atom composition, residue composition, hydrophobicity, polarity, and charge, and (ii) geometrical properties such as the number of residues and atoms per pocket, volume, and sphericity.

Fpocket provides a prediction of pocket druggability. Pockets were defined as druggable if their Fpocket druggability score exceeded 0.5. To visualize the space sampled by the pockets, we performed a principal component analysis (PCA) based on the selected physicochemical and geometric descriptors used to calculate druggability using the Factoextra package in R software.

## 4. Conclusions

Building on recent literature and our group’s previous work suggesting that NS1 may be a promising drug target for influenza infections, we carried out a druggability analysis of the influenza virus’ NS1 protein, particularly of its effector domain (ED). We explicitly considered the intrinsic flexibility of the protein by performing MD simulations coupled with predictions of druggable pockets. This analysis allowed us to identify several druggable pockets on the surface of the ED that is at least partially conserved throughout most of the simulation time. Concomitantly, we carried out an analysis of the stability of the binding region of NS1 to PI3K and TRIM25 by molecular dynamics that identified the most stable regions in the interfaces of these biologically relevant protein–protein complexes. We concluded that some of the pockets that were partially conserved throughout most of the simulation time co-localize with the most stable regions and essential residues of the binding interfaces between NS1 and the host proteins PI3K and TRIM25, which indicates the potential use of these druggable pockets as targets for new antiviral development. 

Therefore, we plan to use these druggable pockets as starting points for the design of a fragment-based drug discovery workflow with the objective of developing novel molecules with antiviral activity against influenza.

## Figures and Tables

**Figure 1 ijms-24-02977-f001:**
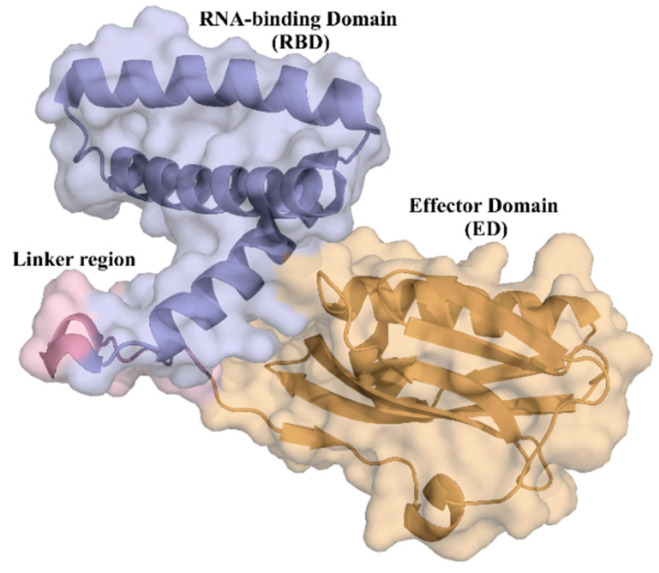
Cartoon depiction of the tridimensional crystal structure of the full-length monomer of NS1 from H6N6 influenza virus with its two structural domains highlighted (Protein Data Bank (PDB) access number 4OPH): the RNA-binding domain (blue) comprising three α-helices; and the effector domain (orange) with α-helices and β-strands. The two domains are connected by a linker region (pink).

**Figure 2 ijms-24-02977-f002:**
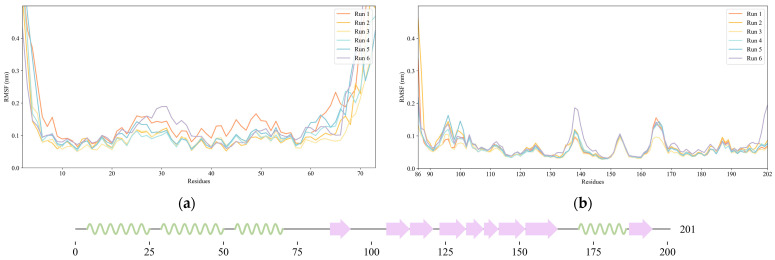
Root mean square fluctuation (RMSF) of NS1 Cα atoms obtained from six independent molecular dynamics simulations. (**a**) RMSF of RNA-binding domain (RBD). (**b**) RMSF of effector domain (ED). A diagram representing the secondary structure of the protein along its amino acid sequence is shown at the bottom of the figure (α-helices represented as green coils and β-strands as pink arrows).

**Figure 3 ijms-24-02977-f003:**
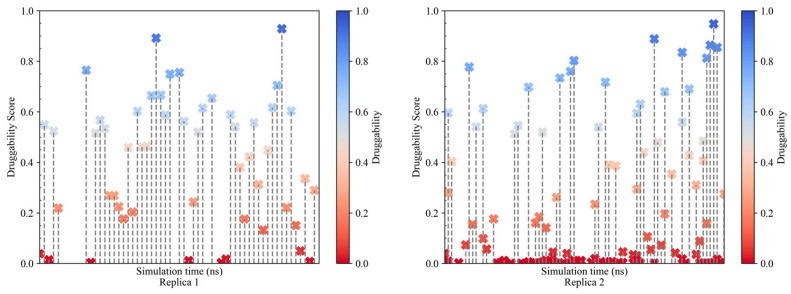
Druggability score of all the pockets identified in full-length NS1 over the sampled conformations obtained for six independent MD simulations.

**Figure 4 ijms-24-02977-f004:**
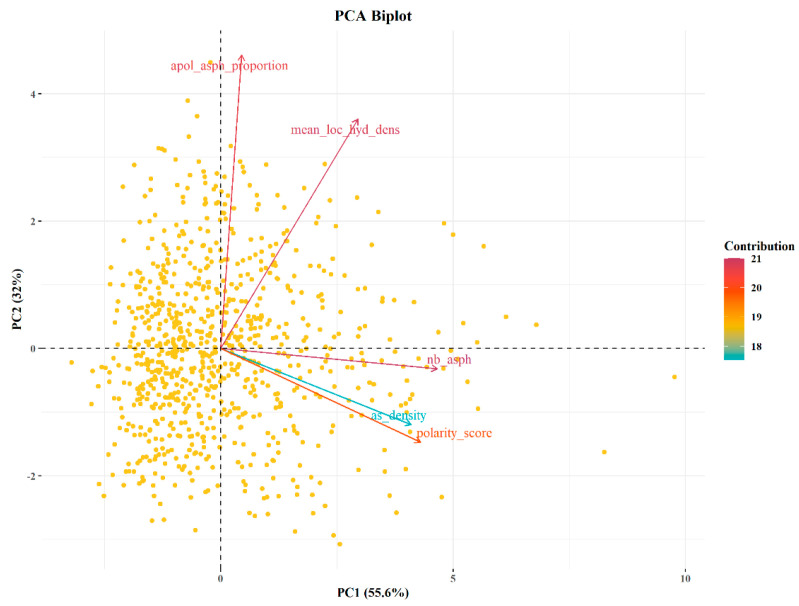
Principal components analysis (PCA) of the physico-chemical properties used to calculate the druggability score with Fpocket.

**Figure 5 ijms-24-02977-f005:**
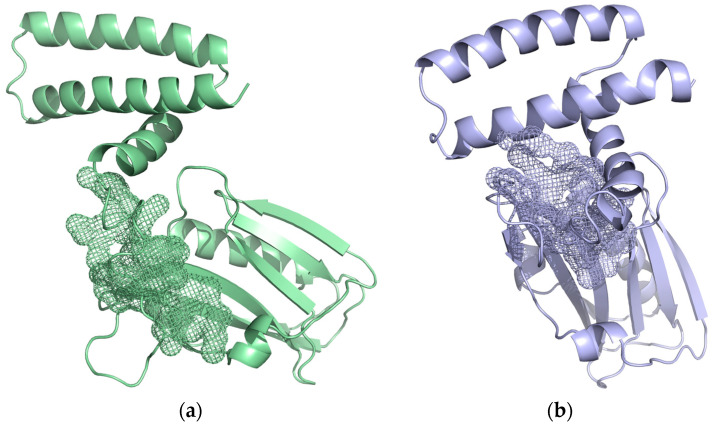
Structure of the NS1 subunit highlighting the pockets with the highest druggability score. (**a**) Snapshot of the 97 ns of run 2 with a pocket in the same color represented in mesh with a druggability score of 0.947. (**b**) Snapshot of the 99 ns of run 3 with a pocket in the same color represented in mesh with a druggability score of 0.946.

**Figure 6 ijms-24-02977-f006:**
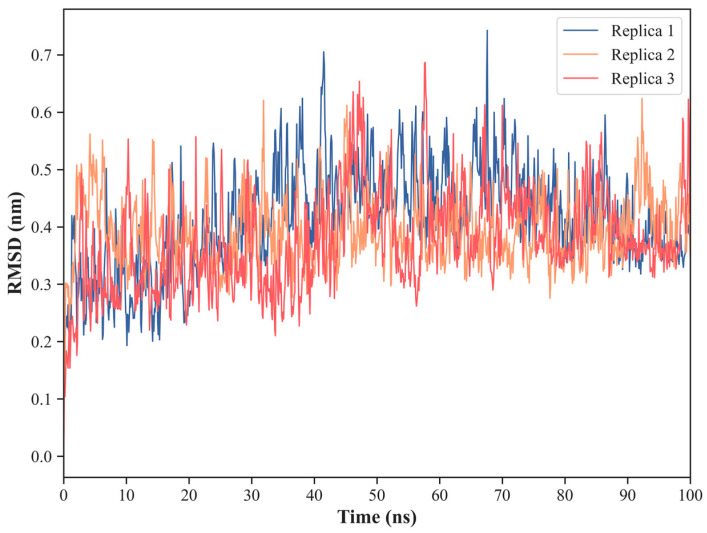
Cα atoms root mean square deviation (RMSD) calculated over the MD simulation trajectories for the complex between the effector domain (ED) of NS1 and the iSH2 domain of the p85β subunit of human PI3K (PDB ID: 6OX7).

**Figure 7 ijms-24-02977-f007:**
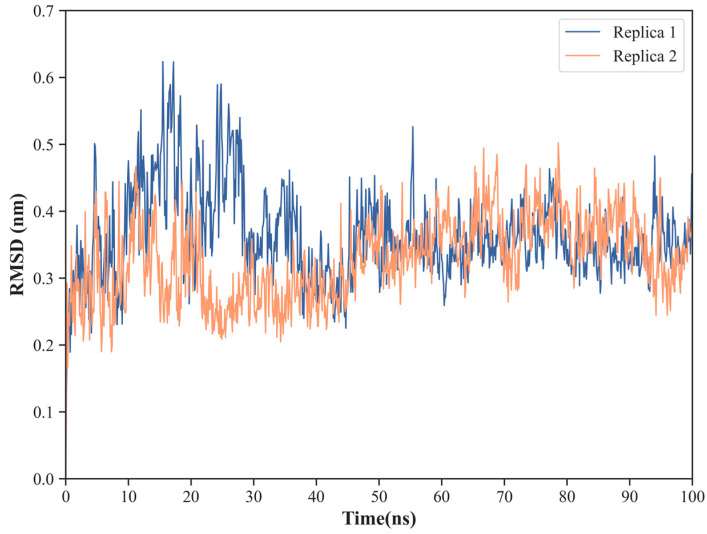
Cα atoms RMSD calculated over the MD trajectories for the complex between NS1 and the coiled-coil domain of human TRIM25.

**Figure 8 ijms-24-02977-f008:**
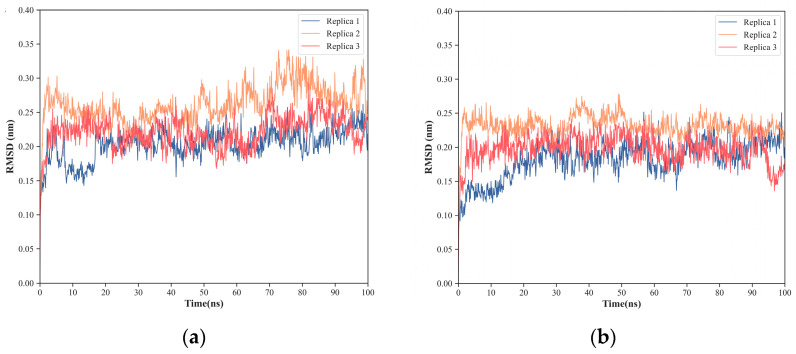
Cα atoms RMSD for the residues in the interface of the complex between NS1 and PI3K, considering the residues from both NS1 and PI3K (**a**) or just the residues from NS1 (**b**).

**Figure 9 ijms-24-02977-f009:**
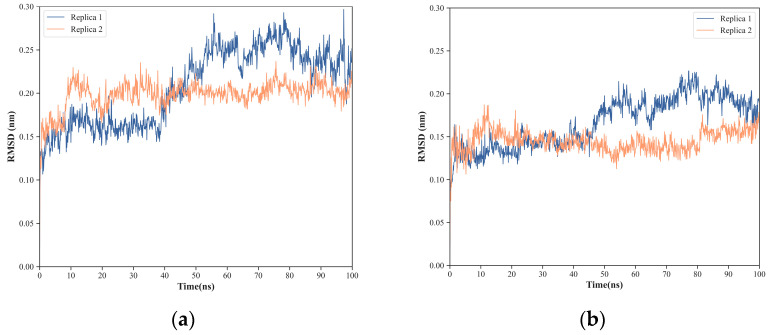
Cα atoms RMSD for the residues in the interface of the complex between NS1 and TRIM25, considering the residues from both NS1 and TRIM25 (**a**) or just the residues from NS1 (**b**).

**Figure 10 ijms-24-02977-f010:**
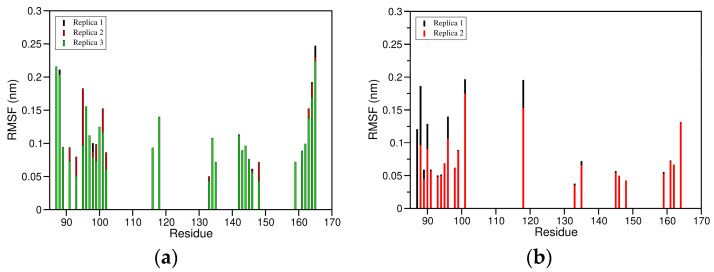
RMSF of NS1 interfacial residues. (**a**) RMSF of the NS1 interfacial residues with PI3K; (**b**) RMSF of the NS1 interfacial residues with TRIM25.

**Figure 11 ijms-24-02977-f011:**
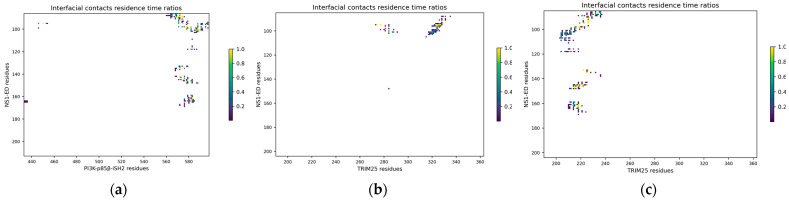
Intermolecular contact maps of the complexes between (**a**) NS1 and PI3K; (**b**) NS1 and the chain I of TRIM25; and (**c**) NS1 and the chain N of TRIM25.

**Figure 12 ijms-24-02977-f012:**
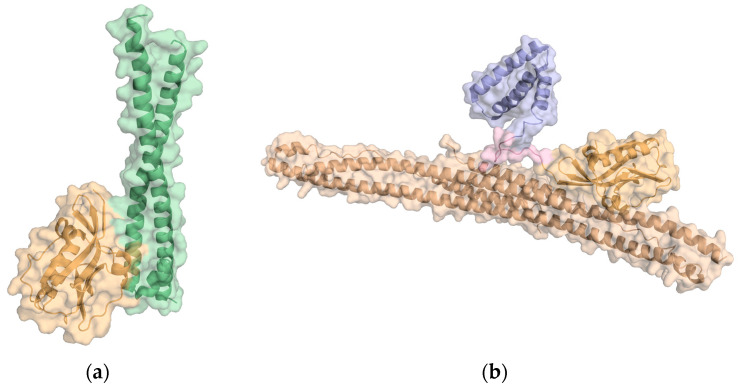
Cartoon representation of (**a**) crystal structure of the complex between ED-NS1 and p85β subunit of PI3K (PDB ID: 6OX7); (**b**) crystal structure of the complex between NS1 and the coiled-coil domain of TRIM25 (PDB ID: 5NT2).

**Table 1 ijms-24-02977-t001:** Amino acid residues constituting the interfacial region in the NS1-PI3K complex.

NS1 Residues	PI3K Residues
Ser 87	His 450
Arg 88	Met 568
Tyr 89	Arg 571
Thr 91	Lys 572
Met 93	Arg 574
Leu 95	Asp 575
Glu 96	Gln 576
Glu 97	Leu 578
Met 98	Val 579
Ser 99	Trp 580
Arg 100	Thr 582
Asp 101	Gln 583
Trp 102	Lys 584
Cys 116	Ala 586
Arg 118	Arg 587
Asn 133	Gln 588
Phe 134	Lys 589
Ser 135	Ile 591
Asn 133	Asn 592
Phe 134	Leu 595
Ser 135	
Glu 142	
Thr 143	
Leu 144	
Ile 145	
Leu 146	
Arg 148	
Glu 159	
Ser 161	
Pro 162	
Leu 163	
Pro 164	
Ser 165	

**Table 2 ijms-24-02977-t002:** Amino acid residues constituting the interfacial region in the NS1-TRIM25 complex.

NS1 Residues	TRIM25
Ser 87	Val 208 (chain N)
Arg 88	Ser 211 (chain N)
Tyr 89	Gln 212 (chain N)
Leu 90	Asn 214 (chain N)
Thr 91	Gly 215 (chain N)
Met 93	Arg 218 (chain N)
Thr 94	Ala 219 (chain N)
Leu 95	Asp 221 (chain N)
Glu 96	Asp 222 (chain N)
Met 98	Val 223 (chain N)
Ser 99	Asn 225 (chain N)
Glu 101	Arg 226 (chain N)
Arg 118	Asp 229 (chain N)
Asn 133	Met 232 (chain N)
Ser 135	Arg 236 (chain N)
Ile 145	Phe 274 (chain I)
Leu 146	Ile 277 (chain I)
Arg 148	Leu 281 (chain I)
Glu 159	Val 322 (chain I)
Ser 161	Tyr 323 (chain I)
Pro 162	Ile 324 (chain I)
Pro 164	Pro 325 (chain I)
	Glu 326 (chain I)
	Val 327 (chain I)

**Table 3 ijms-24-02977-t003:** Stable pairwise residue–residue interactions at the interface of the complex between NS1 and PI3K.

NS1 Residues	PI3K Residues	Residence Time Ratio
Tyr 89	Met 568	1.00
Tyr 89	Lys 572	1.00
Tyr 89	Asp 575	1.00
Tyr 89	Arg 571	1.00
Ser 135	Lys 572	1.00
Ser 99	Leu 578	1.00
Ser 99	Ile 591	1.00
Met 98	Leu 578	1.00
Met 98	Asp 575	1.00
Leu 146	Val 579	1.00
Leu 95	Leu 595	1.00
Leu 95	Leu 578	1.00
Leu 95	His 450	1.00
Leu 95	Arg 574	1.00
Ile 145	Val 579	1.00
Ile 145	Lys 572	1.00
Ile 145	Gln 576	1.00
Ile 145	Asp 575	1.00
Asn 133	Asp 575	1.00
Thr 91	Arg 571	0.99
Asp 101	Gln 588	0.99
Ser 99	Gln 588	0.98
Met 98	Val 579	0.98
Leu 95	Asp 575	0.98
Arg 148	Thr 582	0.98
Ser 161	Gln 583	0.97
Pro 162	Gln 583	0.97
Arg 100	Gln 588	0.97
Glu 142	Lys 572	0.96
Thr 143	Lys 572	0.95
Pro 162	Val 579	0.95
Tyr 89	Arg 574	0.94
Asp 101	Arg 587	0.93
Thr 91	Asp 575	0.92
Ser 161	Val 579	0.92
Ser 87	Lys 572	0.92
Thr 143	Gln 576	0.90
Ser 99	Leu 595	0.90
Glu 96	Gln 588	0.87
Ser 99	Thr 582	0.86
Arg 88	Met 568	0.85
Met 98	Thr 582	0.84
Asp 101	Lys 589	0.82
Leu 163	Gln 583	0.78
Tyr 89	Gln 569	0.74
Leu 95	Gln 588	0.73
Ser 99	Asn 592	0.72
Asp 101	Ala 586	0.72

**Table 4 ijms-24-02977-t004:** Stable pairwise residue–residue interactions at the interface of the complex between NS1 and TRIM25 chain N.

NS1 Residues	TRIM25 Residues	Residence Time Ratio
Tyr 89	Asp 229	1.00
Tyr 89	Asp 222	1.00
Tyr 89	Asn 225	1.00
Tyr 89	Arg 226	1.00
Thr 94	Arg 226	1.00
Thr 91	Asp 222	1.00
Thr 91	Arg 226	1.00
Ser 161	Arg 218	1.00
Ser 99	Ala 219	1.00
Pro 162	Arg 218	1.00
Met 98	Asp 222	1.00
Met 98	Arg 226	1.00
Met 98	Arg 218	1.00
Met 98	Ala 219	1.00
Met 93	Arg 226	1.00
Leu 146	Arg 218	1.00
Leu 95	Val 223	1.00
Leu 95	Asp 222	1.00
Leu 95	Arg 226	1.00
Ile 145	Asp 222	1.00
Ile 145	Asp 221	1.00
Ile 145	Asn 225	1.00
Ile 145	Arg 218	1.00
Glu 101	Gln 212	1.00
Asn 133	Asp 222	1.00
Asn 133	Arg 226	1.00
Leu 146	Gly 215	0.95
Ser 135	Asn 225	0.94
Arg 88	Asp 229	0.94
Met 98	Gly 215	0.93
Arg 118	Ser 211	0.93
Arg 148	Gly 215	0.87
Leu 163	Arg 218	0.84
Ser 99	Ala 216	0.83
Ser 161	Asn 214	0.82
Pro 164	Arg 218	0.81
Arg 88	Arg 236	0.75
Ser 87	Met 232	0.74
Ile 145	Ala 219	0.74
Leu 146	Ala 219	0.71

**Table 5 ijms-24-02977-t005:** Stable pairwise residue–residue interactions at the interface of the complex between NS1 and TRIM25 chain I.

NS1 Residues	TRIM25 Residues	Residence Time Ratio
Ser 99	Leu 281	1.00
Ser 99	Ile 324	1.00
Leu 95	Pro 325	1.00
Leu 95	Phe 274	1.00
Leu 95	Ile 324	1.00
Leu 95	Ile 277	1.00
Glu 96	Ile 324	1.00
Leu 95	Val 327	0.99
Ser 99	Lys 284	0.97
Leu 95	Leu 281	0.97
Leu 95	Glu 326	0.94
Glu 101	Lys 284	0.83
Glu 101	Lys 320	0.78

**Table 6 ijms-24-02977-t006:** NS1 residues participating in the most stable NS1-PI3K interfacial pairwise interactions.

Residues	Number of Pairwise Residue–Residue Interactions
Y89	6
L95	6
S99	6
M98	4
D101	4
I145	4
T91	2
T143	2
S161	2
P162	2
S87	1
R88	1
E96	1
R100	1
N133	1
S135	1
E142	1
L146	1
R148	1
L163	1

**Table 7 ijms-24-02977-t007:** NS1 residues participating in the most stable NS1-TRIM25 interfacial pairwise interactions.

Residues	Number of Pairwise Residue–Residue Interactions
L95	10
M98	5
S99	5
I145	5
Y89	4
E101	3
L146	3
R88	2
T91	2
N133	2
S161	2
S87	1
M93	1
T94	1
E96	1
R118	1
S135	1
R148	1
P162	1
L163	1
P164	1

**Table 8 ijms-24-02977-t008:** Hydrogen bonds formed at the interface of the NS1-PI3K complex.

Residue NS1-ED	Residue PI3K-p85β	Residence Time (%)
P162	Q583	26.97
Y89	D575	76.22
N133	D575	4.7
T91	D575	0.3
S99	Q588	1.9
S87	K572	14.29
D101	Q588	42.66
D101	Q588	0.3
D101	Q588	4.6
R118	Q583	3.6
D101	R587	20.78
P164	W580	0.3
Y89	R571	0.2
S165	K584	0.2
S161	Q583	1.3
S165	S434	0.1
T143	K572	0.9
S99	N592	0.3
S161	Q583	9.19
E96	Q588	42.46
S99	Q588	22.78
R88	N561	1.4
T91	R571	0.4
S87	Q569	0.1
A86	Q569	0.1
A86	K572	0.1
W102	R587	0.1
E159	Q583	0.3
M98	T582	0.9
E142	K572	6.59
S135	K572	1.5
Y89	M568	2
M93	R571	0.1

## Data Availability

The data presented in this study are available in the article.

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
