# Peer review of "Unveiling New Druggable Pockets in Influenza Non-Structural Protein 1: NS1–Host Interactions as Antiviral Targets for Flu"

_ijms, 2023, doi:10.3390/ijms24032977_

Round 1
Reviewer 1 Report
The manuscript is devoted to the topical topic of the search for drug targets in the NS1 protein of the influenza virus. The topic is undoubtedly relevant. The authors suggest using molecular dynamics simulations to search for druggable pockets in the protein, in the sites responsible for its interaction with important cellular partners. The logical construction of the experiments and the design of the study do not cause any remarks. Presentation of data and explanations also satisfy the reviewer. There are minor comments on the introduction (for example, on the number of proteins in the proteome of the influenza virus), but they are not significant. However, I believe that the manuscript in its current form cannot be published. There are two significant observations supporting this view.
1. The NS1 protein is conformationally labile. Modeling conformation and searching for druggable pockets at 300K molecular dynamics simulation temperature is not an adequate approach. Interaction with cell partners occurs at 310-312K. 10-12K for NS1 are critical for domain mobility.
2. A simulation time of 100 ns is usually not sufficient for a 26 kDa protein to bypass the conformational space. The RMSD trend in Figures 5, 7, 8 shows that only the initial sections of the equilibrium state are taken into account. Even if the simulations are repeated, the experiments cannot ensure the reliability of the conclusions.
The elimination of these two significant shortcomings will allow the publication of the manuscript while maintaining its structure.
Author Response
Reviewer 1
We start by thanking the reviewer for his/her fast and thoroughly revision of our manuscript. We were glad that the reviewer considers that “the logical construction of the experiments and the design of the study do not cause any remarks”, and that the presentation of data and explanations are also satisfactory. Notwithstanding, the reviewer presents two main observations regarding methodological aspects of our work that need to be addressed. Therefore, we present below our responses to the reviewer. The manuscript was revised accordingly. The changes that have been made to address the reviewer comments were highlighted using the “Track Changes” function of MS Word.
- We agree with the reviewer that the most appropriate temperature for conducting molecular dynamics of the NS1 protein is 310 K (the physiological temperature). We stress out that this was indeed the temperature used in the simulations of the monomeric NS1 protein to search for druggable pockets and that the 300 K referred in the Materials and Methods section for these simulations was a typo for which we apologize and that we corrected in the revised version of the manuscript (line 534 on section 3.2). The only molecular dynamics simulations in which we applied a temperature of 300 K were the ones of the complexes of NS1 with host proteins, in which our main aim was not to model domain mobility but predicting which interfacial contacts contribute the most for the stability of the protein-protein interactions. In this situation, when the focus of our analysis is a region that is relatively stable (i.e. doesn’t have high configurational degrees of freedom), contrarily to what happens with the linker region, the main region determining the mobility of NS1 domains, in our view a difference of 10-12 K will hardly have a significant impact on the conformational dynamics of the interface, determined mainly by the physicochemical character of its constituent residues. Furthermore, conducting these simulations at 300 K (room temperature) facilitates the comparison of our predictions with in vitro experimental assays on NS1-host proteins interactions, generally conducted at the room temperature (25 ºC ≈ 300 K).
- We absolutely agree with the reviewer when he/she states that 100 ns is usually not sufficient for a protein of the size of NS1 (26 kDa) to extensively sample its conformational space. However, neither the simulations of monomeric NS1 nor the simulations of NS1 with host proteins were conducted with the aim of thoroughly describing the conformational landscape of these systems, for which it would require timescales in the order of µs or more or enhanced sampling techniques. Indeed, our aim with the simulations of monomeric NS1 was to identify potential druggable pockets (including transient or cryptic ones), for which we considered a sampling comprising 6 replicas of 100 ns (600 ns in total) as adequate to obtain a sufficiently large and heterogenous conformational space to sample diverse druggable pockets, although a larger timescale may be beneficial. Similarly, in the simulations of NS1 complexes with the host proteins PI3K and TRIM25, the aim was to predict which interfacial contacts are more stable and may contribute more to the binding, and for that kind of conformational sampling a timescale of 100 ns is usually sufficient, although a larger timescale could be beneficial. The RMSD trends observed in Figure 5, 7 and 8 (Figure 6, 8 and 9 in the revised version) referred by the reviewer show relatively small equilibrium RMSD fluctuations without large conformational changes for the NS1-PI3K complex and for the NS1-PI3K and NS1-TRIM25 interfaces, respectively, which, in our view, is expected from 100 ns equilibrium simulations. Further clarifications on the topic were added on lines 165-173 of section 2.1 and on lines 304-309 of section 2.3.
Reviewer 2 Report
In this work, Cunha et al. studied non-structural protein1 (NS1) in complex with PI3K and TRIM25 to identify potential druggable sites for inluenza antiviral therapy. The authors performed molecular dynamics simulations on these systems and identified stable regions that can be potential druggable sites for future research. Below are my comments that I hope the authors can address:
1) I suggest the authors dig more deeply in the results to make conclusions to readers. For example, bullet points 2.1-2.3 are not very significant. The authors only report what they observed from their analysis without highlighting what we can learn from these analysis. For example, what we can learn from the PCA analysis? Or why it is important to show that N- and C- termini and loops are more flexible which is something we expect to see in simulations. By showing these data, what we can learn about the system or the main point of the manuscript? I suggest the authors further work on the manuscript and improve on this point. Alternatively these can be removed from the manuscript since I do not see a connection between these results and the main point of the manuscript.
2) I suggest the authors clarify more about the logic that a stable region is a more likely druggable site. It seems the authors make their conclusion based on this assumption but it is not clear what this is true. For example, it is known that a cryptic pocket can be a valuable site for targeting but the pocket is not formed in the apo state. In this case one cannot expect such a pocket can be stable in simulations. Instead a dramatic motion will happen to the region since the pocket needs to be formed during ligand binding. It is important that the authors further clarify the applicable domain of their work so that the conclusion can be more clear to readers.
3) page 4 line 151: what does "FL" stand for? Also it will be better if the authors specify simulation length on this line (even though it is introduced later in the manuscript).
4) page 4 line 163: how did these scores obtained? I believe they are from Fpocket but it is better clarified here.
5) page 4 line 167: why using a cutoff of 0.6? Also is it the higher the better? More introduction is needed here.
6) Figure 2: if it is the higher score the better druggable site, then I suggest the authors switch their color code (blue to red, 1.0-0.0) Also, I'm not sure I fully understand this figure. What is the x-axis? Are these scores for one specific site? Please clarify.
7) page 5 line 175: please mention the full name of PCA
8) Figure 3: as I mentioned above, what is the main point of this figure? What we can learn from this figure?
9) page 6 line 205: how is the error bar calculated?
10) page 7 line 218: the significant figures are different from other RMSD values. Please keep consistent throughout the manuscript.
11) page 9 line 240: this is where I feel confused since the conclusion is made based on an assumption which is the more stable the site is the more likely the site is druggable. This may be true but the authors need to explain it. Also in the case of a cryptic pocket, this is not true.
12) page 16 line 353: what function in Chimera was used to mutate the residue? Rotamers? If so, then what library? Dunbrack? Please clarify.
13) page 17 line 433: this is where the authors should mention that some pockets can be small in the apo state (e.g. cryptic pockets) yet they are valuable sites for drug design.
Author Response
Reviewer 2
We start by thanking the reviewer for his/her fast and thoroughly revision of our manuscript. We present below our answers to the reviewer comments. The manuscript was revised accordingly and also to present and explain in a clearer way the aims and results of the article. The changes that have been made to address the reviewer comments were highlighted using the “Track Changes” function of MS Word.
- We acknowledge that the results on sections 2.1 and 2.3 could have been more deeply discussed. Therefore, we have added some important discussions on these sections, particularly in what regards to the analysis of the mobility of NS1 on the lines 165-169 and 174-189 of the section 2.1 of the revised manuscript, in which we emphasize the high conformational plasticity in what concerns to the spatial relation between the RBD (RNA binding) and ED (effector) structural domains and the high stability of the structural domains observed in the simulations, which is in agreement with experimental evidence. In section 2.2, we show the importance of the PCA analysis by discussing what the features that were predicted to contribute more to the druggability score mean in what regards to the main properties underlying druggability. Indeed, we verified that the number of alpha spheres fitting the pocket and the mean local hydrophobic density are the features that contribute the most to the druggability score, which indicates that the size and the hydrophobicity of the pocket are essential for its druggability, respectively. This discussion was added at lines 245-247 of the section 2.2 of the revised manuscript.
- This is a very pertinent topic that should have been better clarified in the original version of the manuscript. We believe that the stability of a given pocket or region is tendentially an advantage from a drug discovery perspective as it implies that this pocket will exist during a significant amount of time, which increases the likelihood of it binding a drug-like small molecule. However, this is not the only characteristic to be considered in druggability predictions (and not the most important) as the geometric and physicochemical properties of the pockets are essential for their ability to bind drug-like molecules. On the other hand, the native apo structure of a protein is not a single structure but an ensemble of conformationally similar structures. This often-neglected fact implies that a pocket that is present in the crystallographic apo structure is not necessarily the most stable or most druggable pocket populated by the native or native-like ensemble and that a druggable pocket (e.g. a cryptic pocket) may be present in a significant portion of the ensemble (i.e. has some stability) and not in the crystallographic apo structure. This is the reason why we conducted molecular dynamics to analyse the druggability of NS1, which allowed us to identify some druggable pockets that were absent in the crystallographic apo structure but were nonetheless present during a significant amount of simulation time. This is particularly relevant as some of the so-called cryptic pockets are also present in the native or native-like ensembles of the apo structure, as reported in Sun et al. Structure. 2020. We added a clarification on this topic on the last phrase of the section 2.3 on page 14 and on the lines 459-469 of section 2.4 on pages 20-21 of the revised manuscript.
- The definition of the FL acronym as well as the length of the simulations were added on page 4 lines 175-176 of the revised manuscript.
- The presented druggability scores were obtained using the Fpocket algorithm. A clarification was added on page 5 line 209-210 of the revised manuscript.
- The minimum consensus cutoff for a pocket to be considered druggable is 0.5, which corresponds to half of the druggability scale, that ranges between 0 and 1, with 0 corresponding to the lowest possible druggability score and 1 to the highest. Other cutoffs have been proposed in the literature but are largely arbitrary and don’t gather a consensus. A clarification on this issue was added on the first paragraph of section 2.2 (lines 212-215 of the revised manuscript). Similarly, a correction of the druggability cutoff value employed in this work was made from the erroneously stated cutoff of 0.6 to the actual used cutoff of 0.5 as well as a further clarification on this issue on lines 222-224 of the revised manuscript.
- We implemented the change in the color code proposed by the reviewer on Figure 2. The x-axis represents the simulation time for each molecular dynamics replica as was further clarified in the axis label. The scores represented in the plots correspond to all the pockets sampled by NS1 along the molecular dynamics trajectories and not to just one, as it was further clarified in the figure label on the revised version of the manuscript.
- The definition of the acronym PCA was added on line 240 on section 2.2, page 7 of the revised manuscript.
- We address this pertinent concern raised by the reviewer on the response to the reviewer’s first comment.
- The error values depicted in page 6 line 205 represent the standard error of the mean, whose formula is , where SEM stands for standard error of the mean, S for standard deviation and N for sample size. Notwithstanding, in the revised version of the manuscript, we choose to present the standard deviation instead of the standard error of the mean because it gives a better description of the conformational fluctuations of the NS1 molecular complexes over the simulations.
- The standard errors of RMSD and RMSF were replaced for standard deviations in the revised version of the manuscript (lines 285-286, 292-293, 310-312, 314-315 and 342-343 on section 2.3) and all values were presented with the same number of significant figures to ensure consistency.
- This pertinent topic was already discussed in the response to the second comment. Here, we mention that a clarification on this issue was added to the last paragraph of the section 2.3 on page 14 of the revised version of the manuscript.
- The Chimera function used to mutate the residues was the Rotamers tool using the Dunbrack library. This information was added on the line 515 of the section 3.2 of the revised manuscript.
- We understand the focus of the reviewer on the possible existence of pockets that does not have a drug-like character at the crystallographic apo structure but can acquire these characteristics upon ligand binding and/or in other conformations of the native or native-like ensemble (e.g. cryptic pockets). However, our experimental design, as referred previously on the answer to the reviewer’s second comment, partially accounts for this reality as the volume cutoff of 500 Å3 applies to all the conformations sampled in the molecular dynamics trajectories, allowing the identification of many stable or transient pockets with high druggability (e.g. cryptic pockets) that are not present in the crystallographic apo structure.
Round 2
Reviewer 1 Report
Corrections makes it possible to publish the manuscript in its current form.